# γδ T Cells: A Game Changer in the Future of Hepatocellular Carcinoma Immunotherapy

**DOI:** 10.3390/ijms25031381

**Published:** 2024-01-23

**Authors:** Stavros P. Papadakos, Konstantinos Arvanitakis, Ioanna E. Stergiou, Maria-Loukia Koutsompina, Georgios Germanidis, Stamatios Theocharis

**Affiliations:** 1First Department of Pathology, School of Medicine, National and Kapodistrian University of Athens, 11527 Athens, Greece; stavrospapadakos@gmail.com; 2First Department of Internal Medicine, AHEPA University Hospital, Aristotle University of Thessaloniki, 54636 Thessaloniki, Greece; arvanitak@auth.gr; 3Basic and Translational Research Unit (BTRU), Special Unit for Biomedical Research and Education (BRESU), Faculty of Health Sciences, School of Medicine, Aristotle University of Thessaloniki, 54636 Thessaloniki, Greece; 4Department of Pathophysiology, School of Medicine, National and Kapodistrian University of Athens, 11527 Athens, Greece; stergiouioa@med.uoa.gr (I.E.S.);

**Keywords:** γδ T cells, hepatocellular carcinoma (HCC), immunotherapy, immunosurveillance, CAR T cell therapy

## Abstract

Hepatocellular carcinoma (HCC) remains a global health challenge with limited treatment options and a poor prognosis for advanced-stage patients. Recent advancements in cancer immunotherapy have generated significant interest in exploring novel approaches to combat HCC. One such approach involves the unique and versatile subset of T cells known as γδ T cells. γδ T cells represent a distinct subset of T lymphocytes that differ from conventional αβ T cells in terms of antigen recognition and effector functions. They play a crucial role in immunosurveillance against various malignancies, including HCC. Recent studies have demonstrated that γδ T cells can directly recognize and target HCC cells, making them an attractive candidate for immunotherapy. In this article, we aimed to explore the role exerted by γδ T cells in the context of HCC. We investigate strategies designed to maximize the therapeutic effectiveness of these cells and examine the challenges and opportunities inherent in applying these research findings to clinical practice. The potential to bring about a revolutionary shift in HCC immunotherapy by capitalizing on the unique attributes of γδ T cells offers considerable promise for enhancing patient outcomes, warranting further investigation.

## 1. Introduction

### 1.1. Hepatocellular Carcinoma: Understanding, Challenges, and Therapeutic Horizons

In 2020, approximately 906,000 individuals worldwide received a diagnosis of liver cancer, mainly hepatocellular carcinoma (HCC), ranking it as the third leading cause of cancer-related deaths [1,2], accompanied by a 5-year survival rate of merely 18% [3]. HCC mostly affects individuals aged 60–70 years and is more prevalent in men. The causative factors show regional and ethnic variations, mainly attributed to distinct risk factors. Chronic liver diseases arising from hepatitis B (HBV) or C (HCV), alcohol abuse, fatty liver diseases, obesity, diabetes, and certain rare conditions contribute significantly to its development [4]. Predominantly, 80% of cases occur in sub-Saharan Africa and eastern Asia, where chronic hepatitis B and exposure to aflatoxin B1 are primary risk factors. For hepatitis B patients, the incidence correlates with viral load, infection duration, and liver disease severity. Occult hepatitis B virus infection also heightens the risk due to DNA damage induced by virus integration [5]. In the USA, Europe, and Japan, hepatitis C, coupled with excessive alcohol intake, stands as the primary risk factor. The epidemiology of hepatocellular carcinoma exhibits dynamic temporal trends, with Japan and Europe experiencing plateauing or declining incidence, contrasting with the ongoing increase in the USA, predicted to stabilize by 2020 [5]. Concurrently, there is a noticeable rise in the incidence of metabolic dysfunction-associated fatty liver disease (MAFLD)/metabolic dysfunction-associated steatohepatitis (MASH)-related liver cancers, now standing as the most prevalent chronic liver diseases globally, impacting approximately 25% of the population [6]. Lifestyle factors, such as obesity, diabetes, alcohol consumption, and genetic variants (PNPLA3, TM6SF2, HSD17B13), are associated with an increased risk of HCC, even in the absence of cirrhosis [7]. Notably, enhancing glycemic control and maintaining a healthy weight are independently linked to a reduced HCC risk [8]. Interestingly, while alcohol-related liver disease is more common in men, women face a higher relative risk of developing HCC [9]. Epigenetic mechanisms significantly influence the progression of HCC [10]. For example, the effectiveness of miR-491 in inhibiting the proliferation of HCC cells is evident, as elevated miR-491 expression correlates with reduced levels of EGFR and STAT3 phosphorylation miR-491 plays a pivotal role in impeding the expansion of liver cancer stem cells (CSCs) by inhibiting the activation of NF-κB [10]. In the context of HCC, a mutational analysis focused on specific regions of the Bax protein, particularly the BH3 domain (residues 59–73) and nearby segments (residues 41–58 and 74–93). Noteworthy mutations were identified, with residue A82 being prominently mutated in HCC, along with other mutation sites observed in various cancer types, such as colorectal adenocarcinoma, uterine serous carcinoma, cutaneous melanoma, and esophageal adenocarcinoma. Bax’s role in cancer is linked to its interaction with Bcl-2 in the apoptotic pathway. Bcl-2, an antiapoptotic protein, hinders cell death and promotes tumors. The mechanism of inhibition includes the prevention of cytochrome c release from mitochondria or its binding to Apaf-1, a crucial step in averting apoptosis [11]. Looking ahead, the evolving landscape of molecular- and immune-based therapies holds promise in the advanced treatment of HCC [12,13,14]. As metabolic dysfunction-associated steatotic liver disease (MASLD)/MASH-related liver cancers increase, understanding risk factors and genetic associations as well as developing effective surveillance strategies are critical for managing and preventing this growing health concern [6]. Certain challenges with immunotherapy for HCC limit its therapeutic utility. In the context of the CheckMate 040 trial, the objective response rate (ORR) with the immune checkpoint inhibitor (ICI) nivolumab in patients with HCC was 20% [15]. Only 25–37% of HCCs show an inflammatory response [16]. Montironi et al. have proposed a refined immunogenomic classification [17]. This classification distinguishes HCC into three distinct classes: the inflamed class, the intermediate class, and the excluded class. Despite the promising avenue of immunotherapy, these findings emphasize the complexity of HCC’s immune landscape [18,19,20,21,22,23,24] and the importance of tailoring novel treatments to its diverse molecular features [25].

### 1.2. Decoding γδ T Cell Diversity: Bridging Innate and Adaptive Immunity

Lymphocytes in the vertebrate adaptive immune system utilize somatic recombination to generate diverse antigen receptor repertoires, enabling recognition of a wide range of pathogens. B cells, αβ T cells, and γδ T cells, evolving over approximately 500 million years, are crucial components of this system [26]. While research on T cells has traditionally focused on the αβ compartment, groundbreaking studies revealed the existence of a separate lineage, γδ T lymphocytes, distinguished by a distinct somatically recombined γδ T cell receptor (TCR) and non-MHC restriction [27]. Despite proposing diverse roles in immunity, such as protection against pathogens, tumor immunosurveillance, and the maintenance of epithelial surfaces, the functions and antigen recognition requirements of γδ T cells remain unclear [28].

Regarding human peripheral γδ T cells, two major subsets, innate-like Vγ9Vδ2 T cells and adaptive-like subsets, exhibit distinct immunobiologies [27]. Innate-like Vγ9Vδ2 T cells, predominant in peripheral blood (PB), undergo a phenotypic transition from rare and naïve in cord blood to mature effector cells with CD45RO expression in early childhood [29]. They respond universally to pathogen-derived phosphoantigens (pAgs), including microbial-derived 4-hydroxy-3-methyl-but-2-enyl pyrophosphate (HMBPP), in a TCR-dependent manner [30]. This subset displays a semi-invariant TCR repertoire, resembling innate lymphocytes, such as i-NKT and mucosal-associated invariant T (MAIT) cells. Additionally, Vγ9Vδ2 T cells can be activated in a TCR-independent manner by interleukin (IL)-12/IL-18 stimulation. Their immunobiology is characterized as “innate-like,” encompassing both innate and adaptive features, with potential clonal expansions contributing to functional heterogeneity [29].

On the other hand, adaptive-like subsets, such as Vδ2^neg^ γδ T cells and Vγ9^neg^Vδ2 T cells, constitute a minority in PB [26]. Vδ2^neg^ T cells, enriched in solid tissues, respond to viral infections and display clonal expansions, exhibiting an adaptive-like immunobiology [27]. These cells undergo clonal expansions in response to various infections, including CMV, suggesting an adaptive response [31]. Vγ9^neg^Vδ2 T cells, unable to respond to pAgs, adopt a T_naive_ phenotype, but in rare instances, they can transition to a T_effector_ phenotype with clonal expansion upon stimulation [26]. Both subsets show remarkable parallels with conventional adaptive CD8^+^ T cells, exhibiting clonal expansions, effector transitions, and potential contributions to long-term immunosurveillance [32].

In summary, recent studies reveal a complex landscape of γδ T cell subsets with distinct immunobiologies, combining features of both innate and adaptive immunity [32]. The transition from naïve to effector status in these subsets appears linked to environmental exposures, particularly infections, resembling conventional adaptive immune responses [26]. γδ T cells possess unique characteristics that make them well suited for recognizing and responding to transformed cells, including HCC. γδ T cells exhibit distinct advantages over conventional αβ T cells. Unlike αβ T cells that mainly recognize peptide antigens presented by major histocompatibility complexes (MHC), γδ T cells recognize a diverse array of non-peptide antigens, including stress-induced molecules and phosphorylated metabolites. γδ T cells provide a rapid response to antigens, acting as a frontline defense in epithelial tissues and mucosa. Their innate-like immune functions enable immediate responses without prior sensitization, contributing to immune surveillance against infections and tumors. Importantly, γδ T cells can recognize antigens independently of MHC presentation, broadening their capacity to detect various stress-related molecules. In contrast, αβ T cells primarily function as adaptive immune cells, requiring specific antigen recognition and prior activation [33]. Additionally, allogeneic γδ T cells, characterized by innate cytotoxicity and tissue tropism, present a potential off-the-shelf CAR T cell therapy for solid tumors, including HCC, with low risk of graft-versus-host disease (GvHD). However, limited availability and technical complexities in expansion protocols hinder their widespread clinical use [34]. Further research is needed to fully elucidate the mechanisms governing this unconventional adaptive differentiation and its implications for immunosurveillance.

### 1.3. γδ T Cells: Dynamic Modulators of Inflammation in Chronic Liver Diseases

γδ T cells play a crucial role in the establishment and regulation of inflammatory processes, with their involvement varying based on disease etiology and specific subsets [35]. The involvement of γδ T cells in various inflammatory processes is underscored by several studies, with their role dependent on disease etiology and the specific subset engaged. Notably, Vδ1 and Vγ9Vδ2 T cells can adopt a CD103^+^ liver tissue-resident memory (TRM) phenotype, exhibiting higher levels of C-X-C chemokine receptor (CXCR) 6 and CXCR3 compared to circulating γδ T cells in HCC patients [33]. In acute viral HBV, γδ T cells are depleted in blood but accumulate in the liver, potentially driven by CXCR3 and C-C chemokine receptor (CCR)5 [36]. In chronic HCV infection, Vδ1 T cells are enriched in the liver, correlating with increased liver inflammation [37]. In models of acute liver injury, γδ T cells contribute to pro-fibrotic processes through IL-17A production [38]. However, in humans, γδ T cells are more likely to exhibit type 1 effector functions, producing interferon (IFN)-γ and engaging in tumor-killing activities [39]. Following chronic liver injury, γδ T cells recruited to the liver via CCR6 suppress fibrogenesis by promoting hepatic stellate cells (HSCs) death [40].

In liver transplantation for chronic liver disease, Vγ9^+^Vδ2^+^ T cells decrease in circulation and liver sinusoids, particularly in acute-on-chronic liver failure [41]. Chronically activated, these cells exhibit reduced antibacterial effector functions in response to microbial metabolites, possibly contributing to bacterial infections.

In decompensated cirrhosis, bacterial translocation in the peritoneal cavity leads to a comparable fraction of γδ T cells as in the blood, but with a TRM phenotype [34,42]. This phenotype is associated with enhanced local inflammation in response to a bacterial infection. The detection of both migratory and resident γδ T cell populations suggest a possible redistribution between the gut, liver, circulation, and peritoneum [42]. Furthermore, in the portal vein of patients with decompensated cirrhosis, Vδ1 T cells are relatively enriched, hinting at a potential gut mucosal origin [43].

## 2. Exploring Epigenetic Regulation of γδ T Cells: Insights into Development, Function, and Therapeutic Strategies in Cancer Immunotherapy

Epigenetic regulation plays a crucial role in the process of carcinogenesis [44,45,46,47,48]. γδ T cells, positioned at the interface of innate and adaptive immunity, exhibit a unique repertoire that necessitates in-depth investigation. As our comprehension of the epigenetic landscape evolves, the analysis of the molecular pathways regulating the behavior of γδ T cells becomes invaluable for the comprehension of immune regulation [49]. For example, the utilization of single-cell RNA-sequencing technology and immune functional assays provided evidence that bacille Calmette –Guerin (BCG) induces changes in the epigenetic transcriptional programs of γδ T cells at the chromatin level [50] and at histone H3 acetylation at lysine 27 (H3K27ac) [51], enhancing their responsiveness to heterologous bacterial and fungal stimuli, such as lipopolysaccharides (LPS) and Candida albicans [52], characterized by a higher production of TNF and IFN-γ several weeks after the vaccination [52]. Taking a step further, distinct and temporally restricted epigenetic mechanisms regulate the development of TCRαβ and TCRγδ T cells [53]. Chromatin accessibility dynamics exhibit stage specificity, with T cell lineage commitment marked by the GATA binding protein 3 (GATA3)- and B-cell lymphoma/leukemia 11B (BCL11B)-dependent closing of PU.1 sites. Notably, β-selection is characterized by a temporary increase in histone H3 lysine 27 (H3K27me3) without open chromatin modifications. Emerging γδ T cells, originating from common precursors of β-selected cells, exhibit significant chromatin accessibility changes due to strong TCR signaling [53]. Chromatin modifiers, particularly histone-modifying factors, play a significant role in the differentiation of γδ T cells [54]. Specifically, an upregulation of chromatin modifiers is evident during the differentiation trajectories of γδ T cell subsets (γδ T17 and IFN-γ-producing T cells). The late activated molecules during effector differentiation, particularly in the γδ T17 lineage, showed an increased expression of histone-modifying factors, such as histone–lysine N-methyltransferase (Kmt)2a, Kmt2c, lysine demethylase (Kdm)5a, Kdm5b, and the histone deacetylase (Hdac)7 [54]. These modifications are involved in shaping the functional characteristics and phenotypic diversity of γδ T cell subsets. A growing body of evidence suggests that distinct epigenetic patterns at certain gene loci differentiate the IL-17-producing CD27^−^ γδ T cells from the IFN-γ-expressing CD27^+^ γδ T cells [55]. In CD27^−^ γδ T cells, active H3K4me2 marks were found to accumulate at the *Il17a*, *IL17f*, and *Il22* loci, indicating permissive chromatin configurations. However, in γδ CD27^+^ T cells, these marks were notably absent at these loci. Conversely, the *Ifng* locus displayed H3K4me2 marks in both γδ cell subsets, suggesting a shared epigenetic regulation for *Ifng* in both subsets. An additional investigation into histone modifications revealed that acetylation of histone H3 (H3ac) was exclusive to the *Il17a* locus in γδ CD27^−^ cells, while both CD27^+^ and γδ CD27^−^ T cells exhibited H3ac marks at the *Ifng* locus. These epigenetic modifications were associated with differential cytokine gene expression levels. *Il17a* was highly expressed in γδ CD27^−^ T cells, while *Ifng* showed higher expression in CD27^+^ γδ T cells, consistent with the epigenetic patterns observed [55]. Furthermore, the study explored the epigenetic regulation of factors associated with T helper 1 (Th1) and T helper 17 (Th17) differentiation. Distinct epigenetic patterns were identified for Th1 and Th17 factors in γδ T cell subsets, highlighting unique regulatory mechanisms. The transcriptional polarization of Th17 factors was particularly emphasized, indicating a stronger epigenetic and transcriptional regulation of these factors in γδ T cell subsets [55]. A recent study investigated the regulatory mechanisms of the natural killer group 2D (*NKG2D*) gene, a crucial activating receptor expressed by various immune cells, such as NK cells, NKT cells, γδ T cells, and CD8^+^ T cells [56]. Fernández-Sánchez focused on understanding the epigenetic regulation of the *NKG2D* gene in CD8^+^ T lymphocytes and NK cells. They revealed DNA methylation in the *NKG2D* gene in CD4^+^ T lymphocytes and specific T cell lines (Jurkat and HUT78), while the gene remained unmethylated in *NKG2D*-positive cells (CD8^+^ T lymphocytes, NK cells, and the NKL cell line). Additionally, high levels of histone H3 lysine 9 acetylation (H3K9Ac) were associated with the *NKG2D* gene in unmethylated cells. Treatment with the histone acetyltransferase (HAT) inhibitor curcumin reduced H3K9Ac levels in the *NKG2D* gene. This downregulated *NKG2D* transcription and significantly reduced the lytic capacity of *NKG2D*-mediated NKL cells, indicating the importance of histone acetylation in *NKG2D* gene expression and function. The study concluded that epigenetic mechanisms, specifically DNA demethylation and histone H3K9 acetylation, play a crucial role in regulating *NKG2D* expression in different immune cell subsets [56]. Modulating *NKG2D* expression through epigenetic treatments, such as HAT inhibitors, could potentially offer a new therapeutic strategy for various immunotherapeutic approaches involving the *NKG2D* receptor.

The combination of immune checkpoint blockade (ICB) and epigenetic modifiers significantly enhances γδ T cell functions, providing a promising strategy for improving clinical outcomes in immunotherapy [57]. Using 3D melanoma models, it is observed that γδ T cells infiltrated rapidly but exhibited exhausted phenotypes, limiting tumor killing. ICB enhances γδ T cell killing, and epigenetic modifiers (Entinostat, Vorinostat) improve functions by regulating MHC class I chain-related protein A (MICA)/ MHC class I chain-related protein B (MICB) and programmed death-ligand 1 (PD-L1). γδ T cells, serving as innate immune sentinels, infiltrate melanoma spheroids effectively, with early IFN-γ production and enhanced effector functions. However, they display exhausted phenotypes. Combining γδ T cells with anti-programmed cell death protein 1 (PD1)/cytotoxic T lymphocyte associated protein 4 (CTLA-4) monoclonal antibodies (mAbs) enhanced functions and reduced spheroid size. Epigenetic modifiers such as Tubastatin-A, Ricolinostat, and Vorinostat inhibit HDAC6/7 pathways, improving γδ T cell antitumor functions. These modifiers increase MICA/MICB expression and decrease NKG2A expression in tumor cells. In conclusion, spheroid/melanoma patient-derived organoid (MPDO) models serve as valuable platforms, with ICB and epigenetic modifiers enhancing γδ T cell therapeutic efficacy [57].

## 3. Insights into the Utilization of γδ T Cells in HCC Immunotherapy Clinical Evidence

HCC treatment necessitates innovative immunotherapeutic strategies. The HCC tumor microenvironment (TME) is acknowledged for its immunosuppressive characteristics, attributed to various regulatory factors, such as a low pH, hypoxia, a nutritional deficiency, metabolic remodeling, and inflammation [22]. γδ T cells have emerged as potential assets in this landscape, and their role is currently under intense investigation in patients with chronic liver disease [34]. Emerging evidence suggests that γδ T cells in humans may harbor significant prognostic potential in the context of therapeutic management. 

HCC transcriptomic data reveal a significant upregulation of inhibitory checkpoint molecules in HCC tissues compared to normal tissues [58]. Notably, these molecules, including CTLA4, hepatitis A virus cellular receptor 2 (HAVCR2), lymphocyte-activation gene 3 (LAG3), programmed cell death protein 1 (PDCD1), programmed cell death 1 ligand 2 (PDCD1LG2), T cell immunoreceptor with immunoglobulin (Ig) and immunoreceptor tyrosine-based inhibitory motif (ITIM) domains (TIGIT), and sialic acid-binding Ig-like lectin 15 (SIGLEC15), show consistent expression across different patient pTNM stages, suggesting a uniform responsiveness to ICB therapy. Hu et al. uncovered a distinct immunosuppressive profile in the HCC TME. This is characterized by increased infiltration of regulatory T cells (Tregs), activated mast cells, and M0 macrophages, coupled with the suppression of γδ T cell infiltration and hindrance in the differentiation of macrophages from M0 to M2. Strikingly, these immune cell infiltrations exhibited no statistical differences among different pTNM stages, affirmin g the highly immunosuppressive nature of the HCC TME across various disease stages. The authors further extended their investigation to the phenotypic profiles of circulating γδ T cells in HCC and healthy populations [58]. The analysis reveals a dominance of the Vδ2^+^ subset in the PB of healthy individuals, while under immune-suppressed conditions in HCC, the Vδ2^+^ subset is significantly depleted. The proposed Vδ1^+^/Vδ2^+^ ratio emerges as a potential indicator for clinical prognosis, with higher ratios associating with more suppressed immunity and poorer outcomes. The study also identified reduced *NKG2D* expression in the Vδ1^+^ population, suggesting depressed cell activation and enhanced survival ability, while a higher pPD1 expression in the Vδ2^+^ population indicated suppressed cytotoxicity [58]. These results are in alignment with the research of Wang et al., who aimed to analyze γδ T cell infiltration in HCC for prognosis and therapeutic prediction [59]. Employing the CIBERSORT algorithm and weighted gene co-expression network analysis (WGCNA), they identified γδ T cell-specific genes associated with immunological activity and notably T cell activation. A risk signature comprising 11 hub genes demonstrated robust prognostic capabilities. This signature exhibited correlations with clinical features and tumor grade and stage, providing valuable prognostic insights. High-risk HCC samples exhibited an immune-activated phenotype and increased susceptibility to ICB pathways. The risk signature demonstrated associations with chemotherapy drug sensitivity, suggesting its potential utility in guiding treatment strategies. They also investigated the role of the Rieske Fe-S domain-containing (*RFESD*) gene in HCC, identifying it as a poor prognosis predictor and correlating it with immune cell infiltration and HCC cell line proliferation. Collectively, they underscored the significance of immune infiltration of γδ T cells in the context of HCC, while the constructed risk signature may serve as a valuable tool for risk stratification and treatment decision making in HCC [59]. Analogously, Zhao et al. investigated the role of γδ T cell infiltrates in HCC and their potential correlation with chemokine C-C motif ligand (CCL) and CCL5 expression, along with their impact on 247 Chinese HCC patients’ survival [60]. Their γδ T cell signature was positively correlated with the expression of natural killer (NK) cell receptor genes, indicating heightened T cell-mediated cytotoxic activity. Moreover, a positive correlation was identified between the γδ T cell-specific gene expression and the expression of CCL4/CCL5 and CCR1/CCR5, which are receptors for γδ T cells. Additionally, they provided evidence of the CCL4/CCL5-mediated recruitment of γδ T cells in both in vitro experiments and a murine orthotopic Hepa1–6 HCC model [60]. In conclusion, the study suggested that CCL4/CCL5 may interact with their receptors, CCR1/CCR5, facilitating the recruitment of γδ T cells to tumor regions. This increased the infiltration of γδ T cells in HCC is proposed to enhance antitumor immunity, potentially improving patient prognosis [60].

Sun et al. explored the immune characteristics of hepatitis B virus-related hepatocellular carcinoma (HBV-HCC) and focused on the changes in peripheral immunity in patients with HBV-HCC [61]. They documented that the expression of NKp46, a marker associated with cytotoxicity, is up-regulated in the PB of HBV-HCC patients. Additionally, they investigated γδ T cells, highlighting an increase in PD-1 expression on Vd1 cells in HBV-HCC patients. They suggested a correlation between T cell Ig and mucin domain 3 (TIM3) expression on γδ T cells and disease progression, with increased TIM3^+^ γδ T cells in the PB of advanced HBV-HCC patients. This highlights the need for further exploration of the liver immune microenvironment in HBV-HCC patients, while single-cell RNA sequencing (scRNA-seq) could serve towards a more comprehensive understanding of the pathogenesis and treatment strategies of HCC [61]. Cai et al. investigated the prognostic significance of TCR^+^ γδ T cells in HCC patients following curative resection from a cohort of 342 HCC patients [62]. They reported a diminished presence of γδ T cells in tumoral tissues compared to peritumoral tissues, with peritumoral γδ T cell counts inversely correlating with tumor size. A survival analysis demonstrated that peritumoral γδ T cell levels were associated with time to recurrence (TTR) and overall survival (OS) in a univariate analysis, and with TTR in a multivariate analysis. Notably, peritumoral γδ T cell levels independently predicted TTR in early-stage HCC patients. Conversely, tumoral γδ T cells did not exhibit independent prognostic value. This study concludes that low counts of γδ T cells in peritumoral liver tissue are indicative of a higher risk of recurrence in HCC patients and serve as a predictive marker for postoperative recurrence, particularly in early-stage cases [62]. Finally, Wei et al. explored the role of γδ T cells in predicting HCC recurrence following liver transplantation (LT) [63]. They identified dynamic changes in immune subsets three weeks post-LT, emphasizing increased activated T cells and decreased myeloid-derived suppressive cells (MDSCs). Notably, two immune subsets, CD57^+^ HLA-DR^+^ CD8^+^ T cells and CD28^+^ γδ T cells, showed significant differences between LT recipients with and without HCC recurrence. CD57^+^ HLA-DR^+^ CD8^+^ T cells exhibit a cytotoxic and proliferative phenotype, while CD28^+^ γδ T cells display reduced activation. A tumor-specific CD28^+^ γδ T cell subset with potential antitumor activity is identified, emphasizing its role in preventing tumor recurrence in LT recipients [63].

Evidence suggests γδ T cells play a crucial role in HCC immunotherapy, with studies revealing their potential as prognostic markers and therapeutic targets. The investigations demonstrate their correlation with inhibitory checkpoint molecules, immunosuppressive profiles in the HCC tumor microenvironment, and their predictive value for HCC recurrence post-LT and curative resection.

## 4. The Role of γδ T Cell in the Immune Landscape of HCC

### 4.1. γδ Τ Cell Features and Interactions in the HCC TME

Research has highlighted unique characteristics of γδ Τ cells in the HCC TME. These features determine the interactions between γδ Τ cells and neoplastic cell and the distribution of different γδ Τ cells between the PB and the liver tissue, thus affecting antitumor immunity.

Toutirais investigated the interaction between Vγ9Vδ2 T cells and HCC cells, focusing on the role of DNAM-1 and CD96, two activating NK receptors expressed by Vγ9Vδ2 T cells [64]. The ligands for these receptors, Nectin-like-5, Nectin-2, and CD96, were found on all analyzed HCC cell lines. Through mAb-mediated masking experiments, they demonstrated that DNAM-1 plays a crucial role in the cytotoxicity against HCC cells and IFN-γ production in Vγ9Vδ2 T cells. Specifically, Nectin-like-5, but not Nectin-2, was identified as the ligand responsible for DNAM-1-dependent functions in Vγ9Vδ2 T cells. CD96, however, did not appear to contribute to the killing of HCC cells. Moreover, they revealed that DNAX accessory molecule-1 (DNAM-1) and *NKG2D* could cooperate in the cell lysis of HCC, providing insights into potential therapeutic targets for antitumor immunotherapy involving γδ T cells [64]. Given the liver’s abundance of tissue-resident γδ T cells and the established efficacy of allogeneic Vδ2^+^ γδ T cell adoptive transfer in liver cancer control, Hu et al. investigated the mechanisms underlying γδ T cell repression in the HCC TMΕ [58]. Analyzing circulating γδ T cells in healthy individuals and HCC patients, they observed a significant imbalance in subset proportions. Specifically, the Vδ1^+^ subset exhibited a substantial increase, while the Vδ2^+^ subset showed a marked reduction in the HCC population, resulting in a noteworthy elevation of the Vδ1^+^/Vδ2^+^ ratio. Furthermore, they revealed a significant decrease in *NKG2D* expression within the Vd1^+^ subset, suggesting a potential impairment in antitumor immunity. In contrast, the Vδ2^+^ subset in the HCC group displayed a notable increase in PD1 expression, indicative of compromised cytotoxicity in Vδ2^+^ γδ T cells. Notably, other markers, including *NKG2D*, NKP30, and NKP46, showed no statistical differences between the healthy and HCC groups. These findings shed light on the immunosuppressive nature of the HCC TME, providing insights into the dysregulation of γδ T-cell subsets and potential avenues for therapeutic intervention [58]. This interaction is further supported by Zhao et al., who demonstrated a positive correlation between the γδ T cell signature and the expression of NK cell receptor genes *NKG2D* as well as cytolytic T cell genes (granzymes and perforin), indicating enhanced T cell-mediated cytotoxic activity [60]. They further established a positive correlation between γδ T cell-specific gene expression and the expression of CCL4/CCL5 and their receptors, CCR1/CCR5. In conclusion, they proposed that the interaction between CCL4/CCL5 and their receptors may facilitate the recruitment of γδ T cells to tumor regions. This increased infiltration of γδ T cells in tumors is suggested to enhance antitumor immunity interacting with NK cells, potentially improving the prognosis of HCC patients [60].

### 4.2. γδ Τ Cell Reshaping the HCC Tumor Microenvironment

The immune TME plays a critical role in the dynamics of HCC, influencing its progression and response to treatment [65]. Comprising a complex interplay of immune cells, stromal elements, and signaling molecules, the TME significantly impacts the trajectory of HCC [18,66]. In this context, the immune TME often takes on an immunosuppressive profile, fostering conditions conducive to tumor growth and immune evasion [19,20].

Several T cell populations comprise the immune HCC TME [67]. Within the tumor, an increased presence of cells expressing CD137, and inducible T cell co-stimulator (ICOS) suggested recent antigenic activation. Notably, distinct T cell populations were identified, such as functionally impaired proliferating CD4^+^ cells co-expressing ICOS and TIGIT, functionally active CD8^+^ cells co-expressing CD38 and PD1, and CD4-CD8 double-negative T cell receptor αβ and γδ cells [non-major histocompatibility complex (MHC)-restricted T cells)]. Histologic classification correlated with the accumulation of activated T cells, indicating an immune-inflamed HCC phenotype [67]. Recent findings indicate a synergistic interaction between γδ T cells and NK cells within the HCC TME [60,64,68]. γδ T cells exhibit a capacity to recognize human tumor cells in a non-restricted MHC manner. The mechanisms for malignant cell recognition by γδ T cells involve both TCR-dependent pathways and activation through NK receptors [64]. Toutirais et al. explored the role of two NK receptors, DNAM-1 and CD96, in the lytic function of γδ T cells, as was analyzed previously. These specific interactions between DNAM-1 and Necl-5 on HCC cells was highlighted as a novel regulatory mechanism for γδ T cell cytotoxicity, enhancing IFN-γ production. Moreover, the authors suggested a cooperative role of DNAM-1 and *NKG2D* receptors in triggering the cytotoxic effector function of γδ T cells [64]. Additionally, the expression levels of phospho-antigens in tumor cells play a crucial role in determining the cytotoxicity of hepatic intrasinusoidal (HI) γδ T cells [68]. Kang et al. showed that the cytotoxicity of expanded HI γδ T cells against Huh7 cells was associated with a higher expression of pyrophosphate in Huh7 cells compared to SNU398 cells. In contrast, the cytotoxicity of HI γδ T cells against SNU398 cells depended on the NK receptor *NKG2D*. They also revealed that HI γδ T cells expressed lower levels of PD-1 compared to PB (PB) γδ T cells [68]. Besides phospho-antigens, new proteins, such as hepatocyte growth factor-like protein (also known as macrophage-stimulating protein (MSP)) and peptide HP1, are identified as antigens recognized by γδ T cells in HCC [69]. Furthermore, recent evidence suggests a positive correlation between γδ T cell presence and the expression of NK cell receptor genes, such as *NKG2D*, in tumor transcriptomic data, which was additionally observed in IHC analysis of tumor biopsies from HCC patients [60]. Further research is needed to explore additional NK receptor–ligand pairs in the recognition of HCC and assess their potential as predictive markers for γδ T cell therapy efficacy. Finally, an association between γδ T cells and MAIT cells in the context of the treatment with lenvatinib plus anti-PD1 antibodies in HCC has been reported [70]. Xi et al. demonstrated that tumor necrosis factor (TNF) superfamily member 9 (TNFSF9) is highly expressed in CD8 effector T cells, MAIT cells, and γδ T cells in the treatment group. This suggests that γδ T cells, along with MAIT cells, are part of the immune response activated by the combination therapy. The presence of both γδ T cells and MAIT cells in the treatment group indicates a potential coordinated involvement of these immune cell types in the anti-cancer immune response triggered by the lenvatinib plus anti-PD1 antibody treatment in HCC [70].

Numerous molecules play pivotal roles in shaping the immune TME of HCC, impacting the abundance and functionality of γδ T cells. Emerging evidence suggests that transforming growth factor (TGF)-b1 is associated with a poor prognosis in HCC [71]. Specifically, seven members of the TGF-b family, including bone morphogenetic protein (BMP)2, BMP6, growth differentiation factor (GDF)6, GDF7, GDF10, left-right determination factor 2 (LEFTY2), and TGF-b1, were identified as potential independent prognostic factors for HCC. Jin et al. also demonstrated that TGF-b1 is closely correlated with immune signatures, particularly regulating the immune TME in HCC patients. The authors identified a positive correlation between TGF-b1 expression and regulatory T cells (Tregs) and a negative correlation with γδ T cells [71]. Yi et al. deepened into the interactions between TGF-b1 and γδ T cells in HCC TME [72]. They revealed a significant reduction in the infiltration and cytotoxic function of γδ T cells in HCC tissue. Tumor-infiltrating CD4^+^CD25^+^Treg cells were found to directly suppress the cytotoxic ability and IFN-γ secretion of γδ T cells in vitro, dependent on TGF-b and IL-10. Soluble TGF-b or IL-10, secreted by HCC cells and immune suppressive cells, further directly suppressed γδ T cell activity. Overcoming these immunosuppressive networks was identified as crucial for improving γδ T cell-based immunotherapy [72].

Figure 1 summarizes the interactions between γδ Τ cells and HCC cells as well as with other immune cells responsible for the regulation of γδ Τ cell functions in the HCC TME.

Hu et al. investigated the role of apoptosis, ferroptosis, and pyroptosis in shaping the immunosuppressive environment of HCC [58]. Six genes, including *CASP3*, *GSDME*, *NLRC4*, *NLRP6*, *NOD1*, and *PLCG1*, were found to be correlated with the OS rate of HCC patients, with *GSDME*, *NLRP6*, and *NOD1* significantly correlating with a worse prognosis, particularly in pTNM stage III–IV. These genes were also positively correlated with checkpoint molecule expression (CD27, CD28, CD40, CTLA4, and PD-L1) and immune cell infiltration [58]. They also delved into the imbalance of γδ T cells in the HCC TME. The Vδ1^+^ subset of γδ T cells was significantly elevated, while the Vδ2^+^ subset decreased, leading to an augmented Vδ1^+^/Vδ2^+^ ratio in HCC patients. *NKG2D* expression was significantly reduced in the Vδ1^+^ subset, and PD1 expression is increased in the Vδ2^+^ subset, suggesting suppressed antitumor immunity. Overall, the findings emphasize the collaborative involvement of apoptosis, ferroptosis, and pyroptosis in shaping the immunosuppressive HCC TME and influencing the imbalance of γδ T cells [58].

Summarizing, enhancing the antitumor effect of γδ T cells, and eliminating suppressive factors within the tumor microenvironment is crucial for improving cancer immunotherapy outcomes.

## 5. γδ T Cells: An Immunotherapeutic Odyssey for Hepatocellular Carcinoma

In HCC immunotherapy, γδ T cells show promise for their distinct ability to recognize and target tumor cells, employing mechanisms different from conventional T cells. Their capacity to identify stress-induced molecules on tumor surfaces makes them active contributors to the immune response against cancer. Ongoing research focuses on understanding the multifaceted roles of γδ T cells in the ΤΜΕ and exploring novel therapeutic strategies to enhance cancer treatment efficacy.

### 5.1. Strategies to Enhance γδ T Cell Antitumor Efficacy

#### 5.1.1. γδ T Cells in CAR T Therapies: Targeting HCC Breakthroughs

The development of allogeneic Vδ1 T cells expressing a glypican-3 (GPC-3)-targeted chimeric antigen receptor (CAR) [73] and secreting IL-15 is explored as a potential cell therapy for HCC and other GPC-3-expressing tumors [74]. Autologous CAR T cell therapy has shown success in hematologic malignancies, but challenges exist in treating solid tumors due to issues such as poor T cell homing, CAR target antigen variability, and the immunosuppressive TME. GPC-3 is an attractive target for immunotherapy, being highly expressed in HCC [33,75]. Autologous GPC-3.CAR αβ T cells have shown promise, but logistical challenges and concerns about immune fitness have hindered widespread adoption. Allogeneic cell therapies, derived from healthy donors, present a potential solution by overcoming manufacturing challenges. In fact, recent evidence exploring the safety and efficacy of combining locoregional therapy with the adoptive transfer of allogeneic γδ T cells for advanced HCC and intrahepatic cholangiocarcinoma (ICC) demonstrated that adverse events were manageable, indicating the safety of this novel combination therapy [76], as opposed to allogeneic αβ T cells that may pose a risk of graft-versus-host disease (GvVHD), necessitating complex gene editing [77].

Makkouk et al. investigated the impact of co-expressing soluble IL-15 (sIL-15) on GPC-3.CAR Vδ1 T cells, aiming to enhance their long-term cytotoxicity and proliferation for effective cancer immunotherapy [33]. They expanded PB Vδ1 T cells through selective activation and transduction with a GPC-3-specific CAR and sIL-15. The resulting GPC-3.CAR/sIL-15 Vδ1 T cells demonstrated a less differentiated phenotype, expressing tissue-homing markers and chemokine receptors associated with efficient trafficking to tumor sites. Notably, these engineered cells exhibited robust in vitro antitumor activity against GPC-3-expressing tumor cell lines, even in the presence of soluble GPC-3. Cryopreservation did not compromise their viability and function, suggesting practical clinical applicability. In-depth analyses, including transcriptional profiling and long-term cytotoxicity assays, revealed that sIL-15 significantly contributed to the sustained functional fitness of GPC-3.CAR Vδ1 T cells, promoting durable tumor growth inhibition. Furthermore, in vivo experiments demonstrated the enhanced antitumor activity of GPC-3.CAR/sIL-15 Vδ1 T cells in a xenograft tumor model, showcasing their potential for clinical application without inducing xenogeneic GVHD [33].

Summarizing, the observed tumor-specific activation, efficient control of tumor growth, and absence of xenogeneic GVHD responses in preclinical models support the translational potential of GPC-3 while novel antigens, e.g., hepatocyte growth factor-like protein (MSP) and peptide HP1 have been recognized by γδ T cells in HCC, offering potential candidates for CAR-T therapy [69]. The studies present evidence supporting the potential of allogeneic Vδ1 T cells expressing a GPC-3-targeted CAR with sIL-15 for HCC immunotherapy. The findings include promising preclinical results, such as tumor-specific activation, effective tumor growth control, and the absence of xenogeneic GVHD. Moreover, practical clinical applicability is suggested by the preservation of viability and function during cryopreservation. Given their efficacy in terms of tumor cytotoxicity in xenograft tumor models, future clinical evaluations of allogeneic CAR/sIL-15 Vδ1 T cells in the treatment of GPC-3-expressing tumors, including HCC, is highly expected, since it might offer a safe and effective off-the-self product [33].

#### 5.1.2. Zoledronic Acid/Artesunate and γδ T Cells Reshaping HCC Therapeutics

Ongoing research in the field of HCC investigates the interplay between zoledronic acid (Zol) and γδ T cells [35,78,79,80]. Zol has demonstrated immunomodulatory properties, notably activating γδ T cells, which show antitumor capabilities. The exploration of combining Zol with γδ T cell therapy emerges as a prospective approach to enhance the immune response against HCC.

Zakeri et al. investigated the potential of Zol in enhancing the antitumor activity of Vγ9Vδ2 T cells in the context of HCC with a focus on the TRM subset of Vγ9Vδ2 T cells found in the human liver [35]. They began characterizing the liver-resident Vγ9Vδ2 TRM, highlighting their tissue-homing capabilities through the expression of chemokine receptors CXCR6 and CXCR3. These TRMs displayed a reduced expression of the endothelial homing receptor CX3CR1 and lacked the expression of the lymph node homing receptor CD62L. Functionally, the intrahepatic Vγ9Vδ2 TRMs exhibited an activated profile with higher expression of the T cell activation marker HLA-DR. However, they showed reduced cytotoxic potential, marked by a lower expression of the serine protease granzyme B and a significant capacity for rapid production of the pro-survival cytokine IL-2 and IFN-γ. The authors also explored the antitumor potential of Vγ9Vδ2 TRM against Zol-sensitized HCC cell lines. While fresh intrahepatic Vγ9Vδ2 T cells exhibited minimal effector function in response to co-culture with HCC cell lines, Zol pre-treatment of the tumor cells significantly enhanced the effector function of CD69^+^CD49a^+^ Vγ9Vδ2 TRM. This response was further observed with tumor-infiltrating lymphocytes (TILs) isolated from HCC tumors, suggesting the potential for Zol to boost the antitumor function of intratumoral Vγ9Vδ2 T cells [35]. The above are further supported by Sugai et al., who demonstrated that Vγ9Vδ2 T cells, activated by Zol-induced phosphoantigens, exhibited significant antitumor activity against various HCC cell lines (HepG2, HLE, HLF, HuH-1, JHH5, JHH7, and Li-7). Zol treatment not only increased HCC cell susceptibility to T cell killing but also triggered T cell proliferation and induced cytokine production (IL-4, IL-5, IL-13, IFN-γ, GM-CSF, TNF-α) and grandzyme B [78]. Towards the same direction, Tian et al. utilized Zol in combination with IL-2 in vitro, concluding that, in conjunction, they could expand circulating γδ T cells in HCC patients, enabling them to lyse HCC cells without raising immunosuppressive factors during amplification [80]. Importantly, they investigated immunosuppressive factors, demonstrating that Tregs, γδ T17 cells, and IL-17A do not increase during γδ T cell amplification, indicating the safety of this approach for HCC immunotherapy [80].

Hoh et al. focused on investigating the activity of γδ T cells against pediatric liver tumor cells, specifically hepatoblastoma (HB) and pediatric HCC [79]. They aimed to analyze the ability of ex vivo expanded γδ T cells to recognize and lyse HUH6 and HepT1 cell lines in co-culture assays. The results indicated that incubation of hepatic tumor cell lines with γδ T cells led to a significant decrease in tumor cell viability. Notably, this effect was enhanced by Zol and HDACs. The HDAC inhibitor used was suberoylanilide hydroxamic acid (SAHA or Vorinostat). SAHA inhibited HDACs, specifically HDAC2 and HDAC4, which are found to be upregulated in HB samples. Inhibition of HDACs resulted in the accumulation of acetylated histone H3, a modification associated with open chromatin and enhanced gene transcription. They also introduced MT110, a bispecific BiTE antibody targeting EpCAM/CD3, which significantly enhanced tumor cell lysis by γδ T cells. The effectiveness of γδ T cells was further demonstrated in a spheroid culture model, emphasizing their potential in recognizing and interacting with HB and HCC cells. The optimization of immunotherapeutic strategies, including the use of Zol and bispecific antibodies, holds promise for improving the outcomes of high-risk hepatoblastoma and hepatocellular carcinoma in pediatric patients [79]. Finally, Qian et al. reported that artesunate treatment amplified the cytotoxicity of γδ T cells against HepG2 cells by elevating granzyme B expression. Moreover, it mitigated the inhibitory impact of HepG2 cells on γδ T cells through a reduction in TGF-b1 secretion. Additionally, artesunate boosted the antitumor effect by increasing Fas expression on HepG2 cells, potentially through the inhibition of STAT3 signaling [81]. These findings indicate that artesunate holds promise as a therapeutic intervention for hepatocellular carcinoma.

Concluding, ongoing research explores the synergistic potential of Zol and γδ T cells in HCC therapy. Zol activates Vγ9Vδ2 T cells, enhancing antitumor activity and showing promise in pediatric liver tumors, while artesunate amplifies γδ T cell cytotoxicity against HCC. Emerging evidence supports Zol synergy with γδ T cells for enhanced HCC therapy, activating T cells and sensitizing HCC cells. Pediatric liver tumors show improved outcomes with Zol and immunotherapeutic strategies. Artesunate enhances γδ T cell cytotoxicity against HCC, indicating therapeutic potential.

#### 5.1.3. Therapeutic Manipulation of γδ T Cells in HCC beyond Known Pathways

Yi et al. exploring the immune dynamics within the HCC TME revealed a significant attenuation in the infiltration and cytotoxic function of γδ T cells within HCC tissue [72]. Importantly, Treg cells within the tumor were found to directly suppress the cytotoxic ability and IFN-γ secretion of γδ T cells, with this effect being dependent on TGF-β and IL-10. They emphasized the novel insight into the distribution, cytotoxic function, and Treg-mediated suppression of γδ T cells in the TME. They documented a negative correlation between Treg cells and γδ T cells in HCC tissues. The in vitro analysis revealed a partial reversal of Treg cell-mediated suppression by anti-TGFβ or anti-IL-10 antibodies, suggesting a soluble factor-dependent mechanism in the Treg cell-mediated suppression of γδ T cells [72]. In conclusion, they emphasized the importance of enhancing the antitumor effect of γδ T cells and eliminating suppressive factors, particularly Treg cells, within the ΤΜΕ. The combination of these approaches is suggested as a potential strategy to improve outcomes for patients with HCC.

Finally, Chen et al. investigated the role of miR-382 in sensitizing HCC cells to the antitumor effects of γδ T cells by targeting the expression of cellular FADD-like interleukin-1b-converting enzyme-inhibitory protein (c-FLIP) [82]. HCC tumor tissues and adjacent healthy tissues were collected from patients, and γδ T cells were purified and identified using specific antibodies. The research revealed that ex vivo expanded human γδ T cells exhibited the ability to induce cell lysis of HCC. Notably, miR-382 was found to be downregulated in both HCC tissues and cell lines. They further demonstrated that the overexpression of miR-382 enhanced the sensitivity of HCC cells to γδ T cells. They identified the mRNA of c-FLIP as the specific target of miR-382. Inhibiting c-FLIP with miR-382 significantly promoted the cell lysis of HCC by reinforcing the activation of caspase-8 induced by γδ T cell treatment. In summary, they concluded that the overexpression of miR-382 plays a crucial role in promoting the lysis of HCC cells induced by γδ T cells. This effect is achieved by inhibiting the expression of c-FLIP, highlighting the potential therapeutic significance of miR-382 in enhancing the responsiveness of HCC cells to γδ T cell-mediated immunotherapy [82].

Figure 2 summarizes the proposed strategies applied to enhance γδ Τ cell antitumor efficacy in HCC.

Collectively, recent advances in HCC immunotherapy focus on allogeneic GPC-3.CAR Vδ1 T cells expressing sIL-15, demonstrating tumor-specific activation and effective growth control [33]. Zol and artesunate enhance γδ T cell antitumor capabilities, with Zol sensitizing HCC cells and artesunate boosting cytotoxicity [78]. Therapeutic manipulation addresses Treg-mediated suppression reversal and miR-382 targeting, providing insights into mechanisms for optimizing γδ T cell responses in the HCC TME [82]. These approaches offer promising strategies for improving outcomes in HCC treatment by leveraging the unique characteristics of γδ T cells.

### 5.2. T Cell Exhaustion in γδ T Cells within HCC

In HCC, T cell exhaustion is a complex process influenced by various factors. The liver is an immunologically unique organ, and HCC often develops in the setting of chronic liver inflammation, commonly caused by viral hepatitis or cirrhosis [83,84]. The HCC TME is widely acknowledged for its immunosuppressive nature, influenced by various regulatory factors, such as low pH, hypoxia, nutritional deficiencies, and metabolic pathway alterations [85,86]. The prolonged exposure to antigens in the TME, along with the immunosuppressive signals produced by both the tumor and the liver environment, can lead to T cell exhaustion [87]. Transcriptomic analysis of HCC tissue indicates significantly elevated expression of genes encoding for inhibitory checkpoint molecules, including *CTLA4*, *HAVCR2*, *LAG3*, *PDCD1*, *PDCD1LG2*, *TIGIT*, and *SIGLEC15*, irrespective of patient pTNM stages [58]. A recent surge of evidence regarding PD-1/PD-L1 and LAG-3 has begun to emerge.

More recently, increased evidence highlights the role of the PD-1/PD-L1 axis in the context of γδ Τ cell function in HCC pathogenesis [70,88,89,90]. Jiang et al. documented that circulating γδ T cells from HCC patients exhibit impaired cytotoxicity compared to healthy controls, a finding attributed to a lower frequency of Vδ2 T cells and reduced IL-2 and IL-21 expression [88]. In vitro expansion with Zol enhanced cytotoxicity but did not fully restore it. PD-1 expression was elevated in HCC patients, and a co-culture with γδ T cells increased PD-L1 in HCC cell lines. Blocking PD-1 during the expansion improved cytotoxicity against all HCC lines, while blocking during assays enhanced it specifically against HepG2 and SNU-398, indicating a connection between reduced cytotoxicity and altered IL-2, IL-21, and PD-1 expression in circulating γδ T cells of HCC patients [88]. Ren et al. investigated the impact of IL-35, an immunosuppressive member of the IL-12 family, on HCC by focusing on γδ T cells in the TME [90]. IL-35 stimulation impaired the proliferative and killing abilities of γδ T cells against HCC cells, increased the expression of exhaustion markers (*PDCD1* and *LAG3*) and hampered cytokine secretion. The transcription factor signal transducer and activator of transcription 5a (STAT5A) was upregulated, with a bioinformatic analysis linking it to immune regulatory pathways. A correlation analysis showed a positive association between *STAT5A* expression and tumor immune cell infiltration, *PDCD1* expression, and *LAG3* expression. IL-35 and *STAT5A* were significantly correlated in HCC datasets. In summary, elevated IL-35 levels induce exhaustion and compromise the antitumor function of γδ T cells in HCC [90]. Targeting IL-35 emerges as a potential strategy to enhance γδ T cell-based antitumor therapy for improved prognosis. The long-term use of indomethacin, a potential chemopreventive agent, was explored in HCC [89]. Xu et al., conducting a study using HepA mouse models, revealed that indomethacin exacerbated intrahepatic recurrence, dissemination, and lung metastasis in a dose-dependent manner. Indomethacin inhibited TNF-α and IFN-γ and upregulated PD-1 and PD-L2 expression via TIR (Toll/interleukin-1 receptor) domain-containing adaptor protein inducing interferon beta (TRIF)/nuclear factor kappa-light-chain-enhancer in activated B cells (NF-κB) and Janus kinase (JAK)/STAT3 pathways in γδ T cells. Blocking PD-1 and PD-L2 reversed the reduction in TNF-α and IFN-γ induced by indomethacin. The findings suggest that prolonged indomethacin use contributes to poor prognoses in HCC by promoting PD-1 and PD-L2 expression through TRIF/ NF-κB and JAK/STAT3 pathways, leading to the inhibition of TNF-α and IFN-γ [89].

LAG-3 stands out as one of the primarily studied indicators of T cell exhaustion [90,91]. He et al. investigated the functional state of γδ T cells infiltrating HCC and explored the potential of allogenic Vδ2^+^ γδ T cells in HCC immunotherapy. scRNA-seq on γδ T cells from HCC tumors and healthy donor livers revealed that γδ T cells in the HCC TME exhibit G2/M cell cycle arrest, express cytotoxic molecules, and show functional exhaustion with elevated LAG3 expression. The LAG3^+^Vδ1^+^ population dominated, while Vδ2^+^ γδ T cells were depleted in the HCC TME. Glutamine metabolism was upregulated in γδ T cells due to glutamine deficiency in the TME, resulting in increased LAG3 expression. Ex vivo-expanded Vδ2^+^ γδ T cells from healthy donors complement the loss of TCRclonality and effector functions in HCC-derived γδ T cells [91]. They provided insight into the dysfunctional signatures of HCC-infiltrating γδ T cells and supported the potential use of allogenic Vδ2^+^ γδ T cells in HCC cellular therapy. Fan et al. systematically explored the expression of leukocyte immunoglobulin-like receptor subfamily B (LILRB) on diverse immune cells in HCC, including γδ T cells [92]. LILRB, recognized as inhibitory receptors and potential immune checkpoints, had previously been associated with immune suppression in HCC patients. By analyzing various LILRB family members on immune cells in both PB and the HCC microenvironment, they uncovered specific increases in LILRB expression on γδ T cells among other immune cell types. This unique expression pattern suggests a potential regulatory role of LILRB in modulating the function of γδ T cells within the context of HCC [92]. Further investigation is warranted to understand the functional implications of LILRB on γδ T cells in HCC progression.

Conclusively, several mechanisms contributing to exhaustion in γδ T cells have been identified. These multiple mechanisms collectively contribute to the reduced cytotoxic capacity and functional exhaustion of γδ T cells in HCC. The reversal of the identified mechanisms contributing to γδ T cell exhaustion in HCC presents a unique and promising therapeutic target.

## 6. Discussion

The integrative analysis of HCC over the past decade has refined its classification based on molecular subtyping and correlation with clinical, etiological, and histopathological features [93]. HCCs are initially categorized into two major classes: a “proliferation class” and a “non-proliferation class” [94]. The proliferation class, associated with HBV-related etiology, encompasses clinically aggressive tumors characterized by poor differentiation, frequent vascular invasion, and specific molecular alterations. This class is marked by TP53 inactivating mutations, the amplification of fibroblast growth factor 19 (FGF19) and cyclin D1 (CCND1), and the activation of pro-survival signaling pathways, such as cell cycle, mammalian target of rapamycin (mTOR), RAS/mitogen-activated protein kinase (MAPK), and MET [95,96]. It exhibits chromosomal instability, global DNA hypomethylation, and further subdivision into two subclasses: a “Wnt-TGFb subclass” and a “progenitor subclass” [97]. The “non-proliferation class” of HCC is more heterogeneous and linked to HCV or alcohol-related etiology, including less aggressive, chromosomally stable tumors retaining hepatocyte-like features [98]. This class comprises at least two subclasses, with the well-defined ‘G6’ subclass harboring catenin beta-1 (*CTNNB1*) mutations, leading to greater Wnt/b-catenin pathway activation, telomerase reverse transcriptase (TERT) promoter mutations, and the hypermethylation of genes encoding for cyclin-dependent kinase inhibitor 2A (*CDKN2A*) and E-cadherin (*CDH1*) promoters [98]. CTNNB1-mutated HCCs in this subclass exhibit immunological cold characteristics and specific histological features [16]. The second subclass within the non-proliferation class, named “G4”, is less well defined. It includes a subgroup with a steatohepatitic phenotype and activation of the IL-6/JAK-STAT pathway, identified by the positive expression of C-reactive protein [99]. The G4 subclass further encompasses “polysomy 7” and “interferon” subclasses, involving HCCs with chromosome 7 polysomy and those overexpressing interferon-stimulated genes (ISGs) with an active immune response [93].

HCC progression involves escaping immune surveillance, with checkpoint molecules such as PD-1, CTLA-4, and LAG-3 playing key roles in regulating immune responses [100]. However, not all patients respond to PD-1 or PD-L1 inhibitors in advanced liver cancer, with around 30% showing intrinsic resistance [16,101,102]. Combining PD-1/PD-L1 inhibitors with other agents becomes a rational approach to enhance therapeutic efficacy [13,25]. One strategy involves the dual blockade of CTLA-4 and PD-1/PD-L1, leading to unique immune-stimulating effects [103]. Another strategy targets angiogenesis, a key player in cancer immune evasion. Pro-angiogenic factors inhibit immune cell infiltration and promote T cell exhaustion [25]. Bevacizumab counteracts these effects, potentially enhancing dendritic cell and cytotoxic T cell activation [25]. Tyrosine kinase inhibitors (TKIs) exhibit varying abilities to modulate immune responses, affecting macrophage polarization, T cell function, suppressive cell activity, and immunosuppressive molecule release [104]. However, the effects of TKIs are context-dependent, influenced by dosing and specific TKI types. The complexity of immune responses within the TME has been thoroughly investigated [18,20]. Adding some levels of complexity to this intricate landscape, epigenetic mechanisms, such as DNA methylation, histone modifications, and non-coding RNAs, can also influence the development and progression of HCC by affecting the expression of genes involved in cell growth, differentiation, and apoptosis [105]. This emphasizes the need for repurposing existing medicines [105] and potentially developing new ones to effectively target and modulate the immune responses within the TME [106]. The potential of γδ T cells holds significant promise for advancing therapeutic approaches.

In comparison with other immunotherapies, γδ T cells offer several advantages and disadvantages. In contrast to ICIs targeting inhibitory pathways in αβ T cells, γδ T cells offer an alternative mechanism for tumor recognition and direct killing [34]. With an innate-like recognition of antigens, γδ T cells respond rapidly to stress-induced molecules on tumor cells, potentially overcoming limitations associated with ICI-resistant tumors [34]. This alternative approach to tumor recognition is complemented by the innate and adaptive features of γδ T cells, allowing for direct cytotoxicity without the need for prior sensitization. When compared to CAR T cells, which have demonstrated success in hematological malignancies, γδ T cells may have advantages in solid tumors due to their innate-like recognition and tissue-resident properties [26]. Additionally, γδ T cells can potentially complement cancer vaccines by providing a rapid and targeted response to vaccine-induced antigens [107]. Their ability to offer a versatile and rapid response, along with tissue-resident properties, underscores the potential of γδ T cells in contributing to a comprehensive and effective antitumor immune response, particularly in the context of [107].

The therapeutic transfer of γδ T cells into clinical practice, while holding promise, is not without its limitations. One notable challenge lies in achieving a consistent and robust expansion of γδ T cells ex vivo, a crucial step in generating a sufficient cell population for therapeutic use. Moreover, concerns regarding the persistence and homing ability of transferred γδ T cells within the complex TME of target tissues pose additional challenges. The precise mechanisms regulating the migration and retention of these cells in specific tissues remain areas of research, requiring further elucidation to optimize therapeutic outcomes. Additionally, it is crucial to acknowledge that the clinical data supporting the therapeutic transfer of γδ T cells are currently limited. The majority of evidence stems from preclinical studies, and the transition to comprehensive clinical datasets is in its early stages. Other key challenges include the modest clinical results of the more predominant Vδ2 subset, barriers to developing scaled expansion processes for the less-abundant Vδ1 subset in solid tumors and concerns about cryopreservation impacting cell viability [33]. Addressing these challenges could involve refining strategies for Vδ1 subset expansion, optimizing cryopreservation techniques, and conducting further preclinical and clinical studies to enhance the understanding of γδ T cell therapy dynamics, paving the way for effective and scalable clinical applications. Further comprehensive investigations are warranted to assess the efficacy and safety of epigenetic modifiers such as Tubastatin-A, Ricolinostat, and Vorinostat specifically in the context of HCC. Clinical trials investigating γδ T cell immunotherapy for advanced HCC, such as NCT05628545 (withdrawn due to the COVID-19 pandemic) and NCT04518774 (unknown status), while not conclusive, underscore the evolving landscape of HCC research and the ongoing need for comprehensive data to assess the potential efficacy of γδ T cell-based approaches in human subjects. Furthermore, the heterogeneity of HCC can be considered a significant limitation in the context of γδ T cell-based immunotherapy. This diversity in HCC tumors presents challenges to achieving consistent and robust therapeutic outcomes with this form of immunotherapy. The translation of findings from experimental models to human subjects introduces a level of complexity, and the outcomes observed in preclinical settings may not necessarily mirror the responses seen in diverse patient populations.

Recent research significantly advances HCC immunotherapy by presenting a novel strategy using engineered Vδ1 T cells for HCC treatment and exploring the synergy of Zol and artesunate with γδ T cells. It addresses T cell exhaustion in HCC, uncovering the role of inhibitory molecules such as PD-1 and LAG-3, and proposes potential targets, such as blocking PD-1 and IL-35, for enhancing γδ T cell function. Additionally, it highlights the impact of indomethacin on HCC prognosis and suggests investigating LILRB’s role in modulating γδ T cell function. These findings collectively provide promising avenues for combating γδ T cell exhaustion and improving HCC therapy.

## 7. Conclusions—Future Perspectives

While the therapeutic transfer of γδ T cells holds promise, addressing these limitations is essential to fully unlock their potential for clinical applications. Ongoing research aims to address these challenges and pave the way for more effective and targeted γδ T cell-based therapeutic interventions.

## Figures and Tables

**Figure 1 ijms-25-01381-f001:**
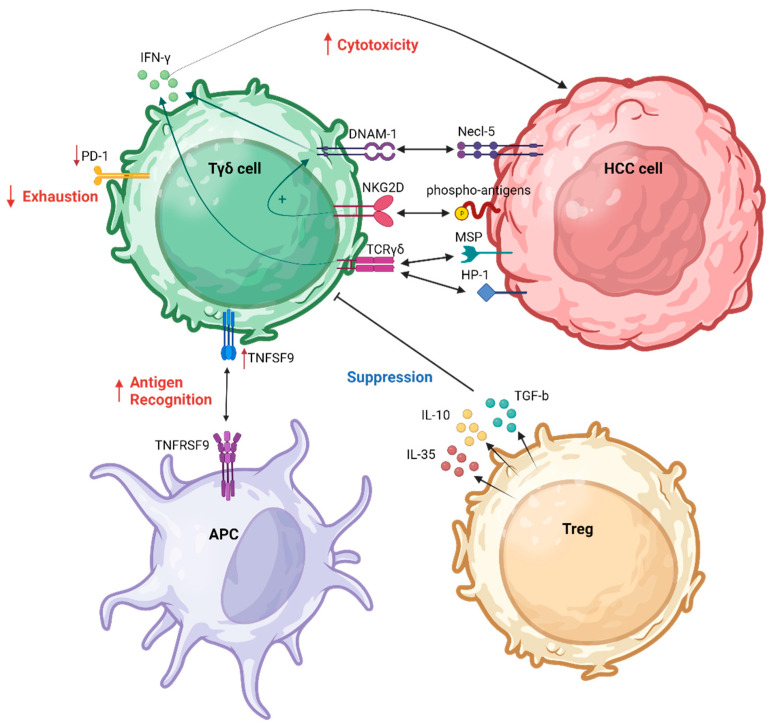
γδ Τ cell interaction in HCC TME. APC, antigen-presenting cell; DNAM-1, DNAX accessory molecule-1; HCC, hepatocellular carcinoma; MSP, macrophage-stimulating protein; *NKG2D*, natural killer group 2D; PD-1, programmed cell death protein 1; TCR, T cell receptor; TNFSF9, tumor necrosis factor (TNF) superfamily member 9; Treg, regulatory T cell. Created with BioRender.com (accessed on 4 December 2023).

**Figure 2 ijms-25-01381-f002:**
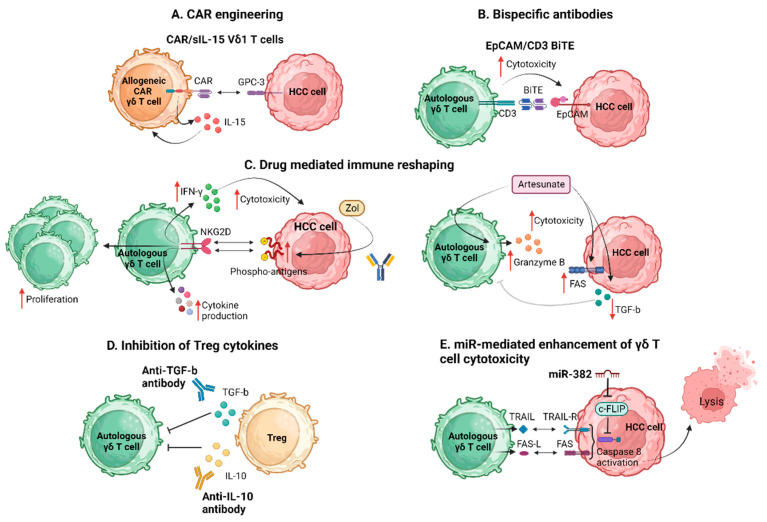
Strategies to enhance γδ Τ cell antitumor efficacy. BiTE, bispecific T cell engager; CAR, chimeric antigen receptor; c-FLIP, cellular FADD-like interleukin-1b-converting enzyme-inhibitory protein; EpCAM, epithelial cell adhesion molecule; FAS-L, FAS ligand; GPC-3, glypican-3; HCC, hepatocellular carcinoma; IFN-γ, interferon-γ; IL, interleukin; miR, micro RNA; *NKG2D*, natural killer group 2D; sIL-15, soluble interleukin 15; TGF-b, transforming growth factor b; TRAIL, TNF-related apoptosis-inducing ligand; Zol, zolendronic acid. Created with BioRender.com (accessed on 4 December 2023).

## Data Availability

Not applicable.

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
