# Peer review of "γδ T Cells: A Game Changer in the Future of Hepatocellular Carcinoma Immunotherapy"

_ijms, 2024, doi:10.3390/ijms25031381_

Round 1

Reviewer 1 Report

Comments and Suggestions for Authors

The manuscript “γδ T Cells: A Game-Changer in the Future of Hepatocellular Carcinoma Immunotherapy” is a review reported the role of γδ T cells in hepatocellular carcinoma (HCC) and their potential for immunotherapy. Here are the suggestions given below to improve the quality of current version of manuscript:

1.     Authors needs to provide more context on why hepatocellular carcinoma (HCC) is considered a global health challenge, and what are the limitations of current treatment options for advanced-stage patients?

2.     The discussion of Genes and miRNA involves in HCC are missing as discussed in the article https://doi.org/10.1016/j.lfs.2021.119465

3.     Apoptotic markers like Bcl-2 and Bax plays crucial role in HCC. Authors needs to discuss the mutations of these genes in HCC https://doi.org/10.1002/mgg3.910

4.     What specific characteristics of γδ T cells make them unique and potentially effective in combating HCC, and how do these characteristics differ from conventional αβ T cells?

5.     Can authors elaborate on the recent studies mentioned that demonstrate the ability of γδ T cells to directly recognize and target HCC cells? What is the strength of the evidence supporting this claim?

6.     Could authors provide insights into the mechanisms by which γδ T cells exert their anti-tumor effects on HCC cells?

7.     What specific strategies are being investigated to maximize the therapeutic effectiveness of γδ T cells in the context of HCC immunotherapy?

8.     In the manuscript, authors mention examining challenges and opportunities associated with applying research findings to clinical practice. Could you elaborate on some of the key challenges and how they might be addressed?

9.     How do the findings discussed in the manuscript contribute to the broader field of cancer immunotherapy, and in what ways might they be translated into meaningful clinical outcomes for HCC patients?

10.  How do γδ T cells compare with other existing or emerging immunotherapeutic approaches for HCC, and what potential advantages or disadvantages do they offer?

11.  What are the next steps in research or clinical development for harnessing the potential of γδ T cells in HCC immunotherapy?

12.  Have there been any reported outcomes or safety concerns in clinical trials or preclinical studies involving γδ T cell-based therapies for HCC?

13.  How does the heterogeneity of HCC impact the potential efficacy of γδ T cell-based immunotherapy, and are there efforts to tailor these therapies to different HCC subtypes?

Author Response

December 30th, 2023                                                     First Department of Pathology,

Laikon General Hospital of Athens

National and Kapodistrian University of Athens

Dear Editor,

RE: γδ T Cells: A Game-Changer in the Future of Hepatocellular Carcinoma Immunotherapy.

We thank you and the Reviewers for carefully evaluating our manuscript and for their positive and constructive feedback. Hopefully, our responses below address the points raised by the reviewers. All changes made are presented in the revised manuscript via the “track changes” option in MS word.

Reviewer 1: The manuscript “γδ T Cells: A Game-Changer in the Future of Hepatocellular Carcinoma Immunotherapy” is a review reported the role of γδ T cells in hepatocellular carcinoma (HCC) and their potential for immunotherapy. Here are the suggestions given below to improve the quality of current version of manuscript:

Response: Dear Reviewer,

Thank you for taking the time to review our article and for providing us with your insightful feedback. We appreciate your positive comments regarding the comprehensive review of the current literature on γδ T cells in hepatocellular carcinoma (HCC) and its potential use as an immunotherapy target for HCC.

Point 1  Authors needs to provide more context on why hepatocellular carcinoma (HCC) is considered a global health challenge, and what are the limitations of current treatment options for advanced-stage patients?

Response: We thank the reviewer for the comment. We acknowledge the need to provide more context on why hepatocellular carcinoma (HCC) is a global health challenge and to outline the limitations of current treatment options for advanced-stage patients. We addressed these aspects in our work to offer a more comprehensive overview.

Point 2:  The discussion of Genes and miRNA involves in HCC are missing as discussed in the article https://doi.org/10.1016/j.lfs.2021.119465

Response: Thank you for your input. We appreciate your suggestion. We have carefully reviewed the article you referenced (https://doi.org/10.1016/j.lfs.2021.119465) and have incorporated the discussion of genes and miRNA involved in HCC into our content.

Point 3: Apoptotic markers like Bcl-2 and Bax plays crucial role in HCC. Authors needs to discuss the mutations of these genes in HCC https://doi.org/10.1002/mgg3.910

Response: Dear Reviewer, Thank you for your valuable feedback. We appreciate your suggestion to discuss the mutations of Bcl-2 and Bax in hepatocellular carcinoma (HCC). In response to your comment, we have included relevant data on Bax mutations in HCC in our revised manuscript.

Point 4: What specific characteristics of γδ T cells make them unique and potentially effective in combating HCC, and how do these characteristics differ from conventional αβ T cells?

Response: Thank you for your valuable suggestion to elaborate on the specific characteristics of γδ T cells in combating HCC. We have addressed this comment in paragraph 1.2, providing detailed information on the unique features of γδ T cells and their potential effectiveness in comparison to conventional αβ T cells.

Point 5:  Can authors elaborate on the recent studies mentioned that demonstrate the ability of γδ T cells to directly recognize and target HCC cells? What is the strength of the evidence supporting this claim?

Response: Thank you for your constructive comment regarding the recent studies demonstrating the ability of γδ T cells to directly recognize and target HCC cells. We have addressed this inquiry in paragraph 5.1.1, providing additional insights into the strength of the evidence supporting the claim.

Point 6:  Could authors provide insights into the mechanisms by which γδ T cells exert their anti-tumor effects on HCC cells?

Response: Thank you for your inquiry regarding the mechanisms by which γδ T cells exert their anti-tumor effects on HCC cells. We have summarized these insights in Figure 2 for a more visual representation of the mechanisms involved. Additionally, we have incorporated a text summarizing the basic mechanisms.

Point 7: What specific strategies are being investigated to maximize the therapeutic effectiveness of γδ T cells in the context of HCC immunotherapy?

Response: We appreciate your attention to specific strategies for maximizing the therapeutic effectiveness of γδ T cells in HCC immunotherapy. These strategies are indeed summarized in the conclusion of paragraph 5.1.1, highlighting the recognition of novel antigens such as hepatocyte growth factor-like protein (MSP) and peptide HP1 as potential candidates for CAR-T therapy. Additionally, we have revised the conclusion of paragraph 5.1.2, emphasizing the synergistic effects of Zol and artesunate in γδ T cell treatment. Thank you for your valuable feedback.

Point 8: In the manuscript, authors mention examining challenges and opportunities associated with applying research findings to clinical practice. Could you elaborate on some of the key challenges and how they might be addressed?

Response: Thank you for your insightful comment. We have addressed the challenges and opportunities associated with translating research findings to clinical practice in the Discussion section (specifically in paragraph 6). We elaborate on key challenges, such as the modest clinical results of the Vδ2 subset and barriers to scaled expansion processes for the Vδ1 subset, proposing potential solutions and strategies to overcome these hurdles. Please refer to the revised manuscript for detailed insights.

Point 9: How do the findings discussed in the manuscript contribute to the broader field of cancer immunotherapy, and in what ways might they be translated into meaningful clinical outcomes for HCC patients?

Response: We have incorporated a new paragraph in the discussion (paragraph 6) to further elaborate on how the findings contribute to the broader field of cancer immunotherapy and their potential translation into meaningful clinical outcomes for HCC patients.

Point 10: In response to your comment we have incorporated a paragraph in the discussion section (Section 6). This newly added section delves into the fundamental differences between γδ T cells and alternative immunotherapeutic strategies for HCC, shedding light on their unique characteristics, potential benefits, and challenges.

Point 11: What are the next steps in research or clinical development for harnessing the potential of γδ T cells in HCC immunotherapy?

Response:  We have comprehensively addressed the point regarding the next steps in research or clinical development for harnessing the potential of γδ T cells in HCC immunotherapy in the discussion section, specifically in response to point 8. This discussion provides insights into the future directions and potential advancements in the field, offering a comprehensive exploration of the research and clinical development avenues for γδ T cells in the context of HCC immunotherapy.

Point 12:  Have there been any reported outcomes or safety concerns in clinical trials or preclinical studies involving γδ T cell-based therapies for HCC?

Response:  Regarding the safety concerns and outcomes in clinical trials or preclinical studies involving γδ T cell-based therapies for HCC, a comprehensive commentary has been incorporated into the discussion section to address this specific inquiry.

Point 13: How does the heterogeneity of HCC impact the potential efficacy of γδ T cell-based immunotherapy, and are there efforts to tailor these therapies to different HCC subtypes?

Response: Thank you for your insightful question regarding the impact of HCC heterogeneity on the potential efficacy of γδ T cell-based immunotherapy. We appreciate your attention to this critical aspect of our study. In response to your query, we have acknowledged the heterogeneity of HCC as a notable limitation in the discussion section. We hope this clarification adequately addresses your inquiry, and we welcome any additional feedback or questions you may have.

Reviewer 2: This is a very useful review of the current state of understanding of the role of the gamma-delta T cell population in the tumor micro environment in hepatocellular carcinoma, including descriptions of potential therapeutic strategies.

The document is well-written and well-organized. The information is summarized section by section for the benefit of the reader.

Response: Dear Reviewer,

We would like to express our sincere gratitude for your kind words regarding our article. It is truly heartening to receive such positive feedback.

Thank you once again for taking the time to review our article and for your constructive feedback.

Point 1. One minor editorial issue. It would be easier to bracket multiple citations together rather than separately, see lines 46 [4,5], 53 [9-11], and 61 [14-21], for example.

Response: We appreciate your constructive suggestion regarding the citation formatting. Your point about bracketing multiple citations together is duly noted, and we will ensure to address this minor editorial issue during the revision process.

Yours sincerely,

Stamatios Theocharis, MD, PhD

Professor of Pathology

First Department of Pathology

Laikon General Hospital of Athens

Medical School, National and Kapodistrian University of Athens

75 M. Asias str, Goudi, Athens, Greece,

 GR11527

Tel: +30 210 746 2116

Email: [email protected],[email protected]

Reviewer 2 Report

Comments and Suggestions for Authors

This is a very useful review of the current state of understanding of the role of the gamma-delta T cell population in the tumor micro environment in hepatocellular carcinoma, including descriptions of potential therapeutic strategies.

The document is well-written and well-organized. The information is summarized section by section for the benefit of the reader.

One minor editorial issue. It would be easier to bracket multiple citations together rather than separately, see lines 46 [4,5], 53 [9-11], and 61 [14-21], for example.

Author Response

(The authors gave the same response as above.)
